# Inferring flow energy, space and time scales: freely-drifting vs fixed point observations

Aurelien Luigi Serge Ponte[1], Lachlan C Astfalck[2,3], Matthew D Rayson[2], Andrew P Zulberti[2], and Nicole L Jones[2]

[1]Ifremer, Université de Brest, CNRS, IRD, Laboratoire d'Océanographie Physique et Spatiale, IUEM, Brest, France
[2]Oceans' Graduate School, The University of Western Australia, Crawley, Australia
[3]School of Physics, Mathematics and Computing, The University of Western Australia, Crawley, Australia

**Correspondence:** Aurélien Ponte (aurelien.ponte@ifremer.fr)

**Abstract.** A novel method for the inference of spatiotemporal decomposition of oceanic surface flow variability is presented and its performance assessed in a synthetic idealized configuration with horizontally divergentless flow. Inference methodology is designed for observations of surface velocity. The ability of networks of surface drifters and moorings to infer the spatiotemporal scales of surface ocean flow variability is quantified. The sensitivity of inference performance for both types of platforms to the number of observations, geometrical configurations, and flow regimes are presented. As drifters simultaneously sample spatial and temporal variability, they are shown to be able to capture both spatial and temporal flow properties even when deployed in isolation. Moorings are particularly adept for the characterization of the flow's temporal variability, and may also capture spatial scales provided they are deployed as arrays. In particular, we show that our method correctly identifies whether drifters are preferentially sampling spatial vs temporal variability. Pending further developments, this method opens novel avenues for the analysis of existing datasets as well as the design of future experimental campaigns targeting the characterization of small scale (e.g. <100 km) ocean variability.

## 1 Introduction

Characterizing oceanic surface motions in terms of their spatial and temporal scales is a recognized pathway toward the identification of the numerous processes that occur in the ocean as well as toward an improved understanding of their occurrences, life cycle, interactions and impact on other components of the ocean variability (Ferrari and Wunsch, 2009). For example, Arbic et al. (2014) relied on horizontal wavenumber-frequency decompositions in order to quantify and rationalize the impact of ocean mesoscale turbulence on longer term ocean variability in idealized, realistic numerical simulations and altimetric observations. At higher frequencies, wavenumber-frequency decomposition enables the separation of internal gravity waves and balanced motions which share similar spatial scales and are therefore entangled in instantaneous two-dimensional data sets (Torres et al., 2019; Jones et al., 2023). For example, using a wavenumber-frequency decomposition, Qiu et al. (2018) were able to quantify the so-called 'transition scale' above which altimetric observations are dominated by balanced turbulence and below which smaller scales are dominated by internal gravity waves. These decompositions are easily performed with numerical simulation output which are provided on complete and regular spatial and temporal grids. However, the lack of ob-

servational knowledge of the high frequency and small scale distribution of energy is a recognized limitation for the validation of tide-resolving kilometer resolution global or basin scale numerical models of the ocean circulation (Arbic et al., 2018; Yu et al., 2019b; Arbic et al., 2022).

The characterization of ocean variability in terms of spatial and temporal scales is also relevant from an operational perspective. The description of an ocean variable's autocorrelation properties is required to map sparse observations via optimal interpolation (Bretherton et al., 1976; Bretherton and McWilliams, 1980). For instance, estimation of surface currents heavily relies on the accurate mapping of altimetric observations which consist of narrow (order 5 to 10 km) geographically and temporally distant tracks (Pujol et al., 2016). The advent of wide swath altimetric (Morrow et al., 2019; Fu et al., 2024) and upcoming current measuring satellite missions introduces novel challenges regarding the mapping of the observed variables and the separation of slower balanced motions and faster internal gravity waves. This has motivated the development of novel strategies for the separation of the signatures associated with both classes of motion. These strategies rely on *a-priori* knowledge of the motions' spatial and temporal scales (Barth et al., 2014, 2021; Ubelmann et al., 2021, 2022).

The in situ characterization of ocean variability at small mesoscale to submesoscale (e.g. <100 km, <10 days) has been a central objective for a number of ambitious experimental efforts over the last decade: LatMIX (Shcherbina et al., 2015); Carthe Consortium (Poje et al., 2014; D'Asaro et al., 2018); OSMOSIS (Buckingham et al., 2016; Yu et al., 2019a); SMODE (Farrar et al., 2020). Estimation of the time-space decomposition of upper ocean variability has resulted from the dense dedicated mooring deployments of OSMOSIS and further highlighted difficulties associated with the Doppler shifting of small-scale structures when observed from fixed platforms (Callies et al., 2020). Such experiments incur significant financial and environmental costs, therefore any optimization in the experimental design and/or improved data analysis strategies are advantageous. Drifters are cheap and experimentally light platforms for the spatial and temporal characterization of ocean variability, but require adequate inference methodologies. This study presents one such methodological development.

The characterization of horizontal and temporal variability of oceanic surface motions from observations represents a challenge that depends on the class of motions of interest, the quantity and nature of observations available, and the lack of a methodology that is both sufficiently versatile to the differing observation platforms and mathematically coherent. Fixed point platforms provide information that is horizontally localized over potentially extended time periods with fine temporal resolution. Such data are conducive to temporal decomposition (Polzin and Lvov, 2011). The tracking of surface and subsurface drifting platforms provide ocean current observations which are also amenable to temporal decomposition (Lumpkin et al., 2017), albeit representing the Lagrangian particle thereby aliasing certain spatial characteristics. At daily to monthly time scales, drifters have enabled characterization of mesoscale eddy variability via inspection of surface current autocorrelation or spectral properties (Zhang et al., 2001; Lumpkin et al., 2002; Veneziani et al., 2004; Sykulski et al., 2016) or rotary wavelet decomposition (Lilly and Gascard, 2006; Lilly et al., 2011). The Global Drifter Program has collected surface current information worldwide for ∼30 years. Recently, the advent of GPS and wider bandwidth satellite communications has enabled high frequency sampling of surface drifter positions and a generation of surface drifter velocity datasets with global hourly coverage (Elipot et al., 2016). Over the last decade, global descriptions of the ocean surface high frequency variability have emerged

(Elipot et al., 2010, 2016; Yu et al., 2019b; Arbic et al., 2022). These descriptions are timely to validate recent kilometer scale tide-resolving basin scale numerical simulations (Arbic et al., 2018).

Satellite observations are well posed to characterize surface ocean spatial variability. The constellation of conventional nadir altimeters provide maps of sea level and surface currents which resolve larger mesoscale motions (Ballarotta et al., 2019). However, spatial and temporal gaps between nadir altimeters impose limitations on the resolvable spatial and temporal resolutions (Ballarotta et al., 2022). Consequently, there are multiple spatial and temporal characterizations of ocean variability which combine altimetry with other in situ datasets, e.g. moorings, XBTs, tomography (Zang and Wunsch, 2001; Wunsch, 65    2010; Wortham and Wunsch, 2014). For smaller spatial scales, ship-based measurement of tracers and currents have informed the estimation of spatial scales of ocean variability (Callies and Ferrari, 2013) but such measurements potentially entangle spatial and temporal contributions to an unclear extent. Drifters are thought to offer promising data for the description of smaller mesoscale and submesoscale variability (Balwada et al., 2016, 2021). Dedicated experiments with deployments of a large number of surface drifters such as that conducted by the Carthe Consortium have provided useful datasets to demonstrate 70    small scale ocean variability despite also highlighting potential biases associated with the horizontally divergent character of the flow at these scales (Poje et al., 2017; Pearson et al., 2019, 2020; Wang and Bühler, 2021).

Here we present a new method for the spatial and temporal characterization of oceanic surface flow variability. To test the method we consider an idealized configuration of ocean variability whose properties and synthetic generation are described in Section 2.1. The novel method for the inference of the flow properties is described in Section 2.3. The inference is then applied 75    to several scenarios of observations in order to explore the performance of the inference relative to the number of observations (Section 3.2), to platform spatial separation (Section 3.1), and, to flow regime (Section 3.4). The results are discussed and conclusions drawn in Section 4.

## 2    Method

### 2.1    Idealized ocean surface flow design

We consider a two-dimensional and time variable flow, described by the sum of rotational and divergent contributions:

$$u = -\partial_y \psi + \partial_x \phi, \tag{1}$$
$$v = \partial_x \psi + \partial_y \phi. \tag{2}$$

where $u$ and $v$ are the zonal (toward positive $x$) and meridional (toward positive $y$) velocities in the respective directions $x$ and $y$, $\psi$ is the streamfunction, $\phi$ is the velocity potential and $\partial_x$ and $\partial_y$ are the partial derivatives in $x$ and $y$. We can describe the 85    second-order behavior of $\psi$ and $\phi$, equivalently, by either their covariance functions or spectral densities. For general random fields $a$ and $b$, defined over $\mathbf{x}$, we define the stationary covariance function as $C_{ab}(\boldsymbol{\tau}) = \langle a(\mathbf{x}_0), b(\mathbf{x}_0 + \boldsymbol{\tau}) \rangle$ where the inner product is given as the covariance inner product $\langle a, b \rangle = \mathrm{E}[(a - \mathrm{E}[a])(b - \mathrm{E}[b])]$. Here, the boldface $\mathbf{x}_0$ and $\boldsymbol{\tau}$ denote a location and distance in $\mathbf{x}$, respectively, in space and time. As stationarity is assumed, covariance is defined only as a function of $\boldsymbol{\tau}$. We define the corresponding spectral density as $S_{ab}(\boldsymbol{\omega})$, where the boldface $\boldsymbol{\omega}$ represents a location in wave-number and frequency

space. As shown by Wiener–Khinchin's Theorem, the covariance function and the spectral density are Fourier pairs, so that

$$C_{ab}(\boldsymbol{\tau}) = \int\limits_{-\infty}^{\infty} S_{ab}(\boldsymbol{\omega})\exp(2\pi i\boldsymbol{\omega\tau})\,\mathrm{d}\boldsymbol{\omega}, \quad \text{and} \quad S_{ab}(\boldsymbol{\omega}) = \int\limits_{-\infty}^{\infty} C_{ab}(\boldsymbol{\tau})\exp(-2\pi i\boldsymbol{\omega\tau})\,\mathrm{d}\boldsymbol{\tau}. \tag{3}$$

Given an assumed parameterisation of $C_{\psi\psi}$, $C_{\phi\phi}$ and $C_{\psi\phi}$, the horizontal velocity auto- and cross-covariances are thus

$$C_{uu} = -\partial_{yy}C_{\psi\psi} - \partial_{xx}C_{\phi\phi} + \partial_{xy}C_{\phi\psi} + \partial_{xy}C_{\psi\phi}, \tag{4}$$

$$C_{vv} = -\partial_{xx}C_{\psi\psi} - \partial_{yy}C_{\phi\phi} - \partial_{xy}C_{\phi\psi} - \partial_{xy}C_{\psi\phi}, \tag{5}$$

$$C_{uv} = \partial_{xy}C_{\psi\psi} - \partial_{xy}C_{\phi\phi} + \partial_{yy}C_{\phi\psi} - \partial_{xx}C_{\psi\phi}. \tag{6}$$

Similarly, given the spectral densities $S_{\psi\psi}$, $S_{\phi\phi}$ and $S_{\psi\phi}$, we define the power and cross-power spectral densities of the horizontal velocities as

$$S_{uu} = l^2 S_{\psi\psi} + k^2 S_{\phi\phi} - kl(S_{\psi\phi} + S_{\phi\psi}), \tag{7}$$

$$S_{vv} = k^2 S_{\psi\psi} + l^2 S_{\phi\phi} + kl(S_{\psi\phi} + S_{\phi\psi}), \tag{8}$$

$$S_{uv} = kl(S_{\phi\phi} - S_{\psi\psi}) - k^2 S_{\psi\phi} + l^2 S_{\phi\psi}, \tag{9}$$

where $k$ and $l$ are horizontal wavenumbers. For our numerical experiment, we derive a purely rotational flow by setting $\phi = 0$ and so, simply, $u = -\partial_y\psi$ and $v = \partial_x\psi$. This leads to the covariance functions $C_{uu} = -\partial_{yy}C_{\psi\psi}$, $C_{vv} = -\partial_{xx}C_{\psi\psi}$ and $C_{uv} = \partial_{xy}C_{\psi\psi}$, and spectral densities $S_{uu} = l^2 S_{\psi\psi}$, $S_{vv} = k^2 S_{\psi\psi}$ and $S_{uv} = -kl S_{\psi\psi}$.

To parameterize the flow we seek either a covariance function or spectral density that satisfies the physical requirements of the streamfunction $\psi$; namely, we require a log-linear decay in the high-frequency/wavenumber of the spectral density. A good candidate for this is the isotropic Matérn covariance function (Rasmussen and Williams, 2005) with auto-covariance function and power spectral density

$$C(\boldsymbol{\tau}) = \frac{2^{1-\nu}}{\Gamma(\nu)}(\lambda\|\boldsymbol{\tau}\|_2)^\nu \mathcal{K}_\nu(\lambda\|\boldsymbol{\tau}\|_2), \quad \text{and} \quad S(\boldsymbol{\omega}) = \frac{c_\nu}{\left(\|\boldsymbol{\omega}\|_2^2 + \lambda^2\right)^{\nu + \mathrm{D}/2}}, \quad \text{where} \quad c_\nu = \frac{2^{\mathrm{D}}\pi^{\mathrm{D}/2}\lambda^{2\nu}\Gamma(\nu + \mathrm{D}/2)}{\Gamma(\nu)},$$

$\|\cdot\|_2$ denotes the Euclidean norm/distance, D is the dimension of $\boldsymbol{\tau}$ and $\boldsymbol{\omega}$, $\Gamma(\cdot)$ denotes the Gamma function and $\mathcal{K}_\nu$ is the modified Bessel function of the second kind of order $\nu \geq 0$. For positive half-integers of $\nu$, i.e. $\nu = p - 1/2$ where $p \in \mathbb{N}^+$, $\mathcal{K}_\nu$ has an analytical expression, otherwise it must be numerically calculated. We assume $\psi$ to follow a separable Matérn process in space (D = 2) and time (D = 1), so that $C_{\psi\psi}(\boldsymbol{\tau}) = \Psi^2 C_{ss}(\tau_d)\cdot C_{tt}(\tau_t)$ where $\Psi$ is the standard deviation of the streamfunction, $\boldsymbol{\tau} = [\tau_d, \tau_t]$ where $\tau_d$ represents the isotropic distance in space and $\tau_t$ represents the time-lag, and both $C_{ss}(\tau_d)$ and $C_{tt}(\tau_t)$ are specified as correlation functions, that is, $C_{ss}(0) = C_{tt}(0) = 1$. For the kernel defined over space $C_{ss}(\tau_d)$ we define the slope and decorrelation parameters $\nu_s$ and $\lambda_s$, respectively. For the kernel defined over time $C_{tt}(\tau_t)$. we define the slope and decorrelation parameters $\nu_t$ and $\lambda_t$, respectively. This separability assumption is a concession on realism which substantially eases the computational cost of the flow generation step and is not expected to affect our evaluation of the inference performance

(Wortham and Wunsch, 2014; De Marez et al., 2023). The covariance functions with respect to $u$ and $v$ are thus

$$C_{uu}(\boldsymbol{\tau}) = -\Psi^2 C_{tt}(\tau_t) \cdot \frac{y^2 C''_{ss}(\tau_d) + x^2 \tau_d^{-1} C'_{ss}(\tau_d)}{\tau_d^2}, \tag{10}$$

$$C_{vv}(\boldsymbol{\tau}) = -\Psi^2 C_{tt}(\tau_t) \cdot \frac{x^2 C''_{ss}(\tau_d) + y^2 \tau_d^{-1} C'_{ss}(\tau_d)}{\tau_d^2}, \tag{11}$$

$$C_{uv}(\boldsymbol{\tau}) = \Psi^2 C_{tt}(\tau_t) \cdot \frac{xy \left( C''_{ss}(\tau_d) - \tau_d^{-1} C'_{ss}(\tau_d) \right)}{\tau_d^2}, \tag{12}$$

where primes denote derivatives with respect to the horizontal distance $\tau_d$.

## 2.2 Synthetic flow data generation

The amplitude of the streamfunction $\Psi$ is related to the flow standard deviation $U$ via $\Psi = U \lambda_s \sqrt{(\nu_s - 1)/\nu_s}$. The reference flow simulation is defined such as to be representative of moderately energetic mesoscale turbulence with $U = 0.1$ m/s, $\lambda_s = 100$ km, $\lambda_t = 5$ days (Fig 1) (Ferrari and Wunsch, 2009). Matérn slope parameters are chosen to be $\nu_t = 1/2$ and $\nu_s = 5/2$ leading to a $-2$ temporal spectrum slope and a spatial isotropic spectral slope of $-6$. While the temporal spectral slope fits expectations, the spatial spectral slope is steeper by a value of one (20%) compared to the value typical of quasi-geostrophic turbulence (Callies and Ferrari, 2013; Wortham and Wunsch, 2014). This concession to realism was made because it yields an analytical form for the Matérn covariance function which alleviates the computational cost of the inference substantially. We reparameterize the covariance functions by $\Psi = \gamma \lambda_s$, where $\gamma$ is interpreted as the amplitude parameter of the horizontal velocity process; as well as interpretability, this has some computational benefits.

With the previous choice of parameters, the streamfunction is generated over a 1000 km by 1000 km domain with 2 km grid spacing and over 100 days with hourly resolution (Fig 1). This resolution is a factor of $\sim 50$ times smaller than decorrelation which is considered enough to resolve the synthesized variability and mitigate numerical interpolation errors in Lagrangian numerical simulations. Sizes of the spatial domain and the time series are $\sim 10$ and $\sim 20$ times larger than decorrelation scales which ensures we are capturing multiple, effectively independent, realizations of the process.

The hourly synthetic flow is fed to the Parcels python library configured with fourth order Runge-Kutta time-stepping and the default A-grid interpolation scheme in order to produce synthetic drifter trajectories (Delandmeter and van Sebille, 2019). Drifters are released initially at all flow grid points with the exception of a 20 km strip around all boundaries, amounting to a total of 9216 drifters for each simulation. Trajectories reaching domain boundaries are de-activated and not advected further in time and discarded from the list of observations that will be used for inference. The fraction of trajectories discarded was 52% in the reference configuration. Drifter positions are stored at hourly resolution and velocities estimated from drifter positions with second-order finite differencing. An example of drifter trajectories is shown in Figure 1. The flow amplitude averaged over time and space is about 1.8% larger than that computed from drifter trajectories which reveals small turbophoresis, i.e., concentration of drifters in areas of lower energy (Freeland et al., 1975).

A non-dimensional parameter used to characterize flow is $\alpha = U \lambda_t / \lambda_s$. This parameter is expected to control the relative importance of spatial vs temporal variability in the projection onto Lagrangian time series (Middleton, 1985). In the reference

scenario, the value of $\alpha$ is $0.4$ which is in the range of observed ocean values (Lumpkin et al., 2002). In order to obtain mooring and drifter time series with different $\alpha$ values, the synthetic flow is simply rescaled and new Lagrangian trajectories are simulated with the rescaled flow.

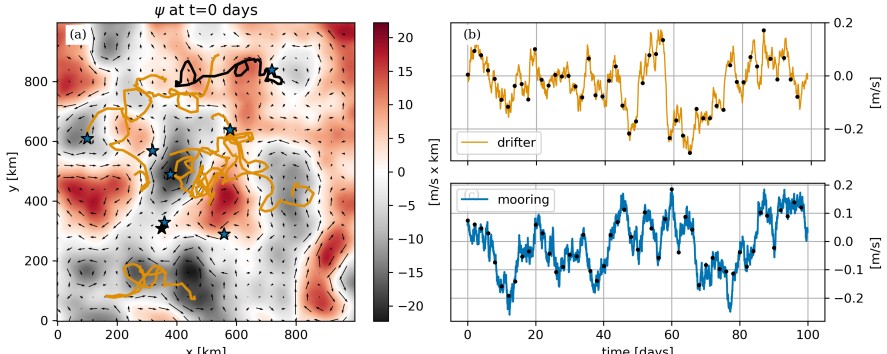

**Figure 1.** Overview of the inference input data for the reference scenario: (a) streamfunction snapshot in color overlaid with eight drifter tracks and moorings used for the inference; (b) x velocity time series of the drifter identified by the black track in (a); (c) x velocity time series at the mooring indicated by the black star in (a). On (b) and (c), black dots indicate the 2 days of sub-sampled data used in the inference.

## 2.3 Inference

Observed data $\mathbf{y}$ is composed of flow time series collected over time by $N_p$ drifters or moorings to which a white noise $\mathbf{n}$ of standard deviation $\sigma$ is added. The critical difference between drifter and mooring observations is that they are collected along drifter trajectories in the former case, i.e. $\mathbf{u}[\mathbf{x}(t)] + \mathbf{n}(t)$ where $\mathbf{x}(t)$ is a drifter trajectory, while they are collected at a fixed location in the latter one, i.e. $\mathbf{u}[\mathbf{x}, t] + \mathbf{n}(t)$ where $\mathbf{x}$ is a mooring location.

We treat the collection of parameters $\Theta = \{\gamma, \lambda_s, \lambda_t, \sigma^2\}$, as uncertain and unknown and probabilistically quantify this uncertainty. We treat $\Theta$ as a random variable and so naturally adopt the Bayesian paradigm of probability. Bayes' Theorem states $\mathrm{p}(\Theta \mid \mathbf{y}) \propto \mathrm{p}(\mathbf{y} \mid \Theta)\mathrm{p}(\Theta)$, where $\mathrm{p}(\Theta \mid \mathbf{y})$ is the posterior distribution, $\mathrm{p}(\mathbf{y} \mid \Theta)$ is the likelihood and $\mathrm{p}(\Theta)$ is the prior distribution. The posterior is our target quantity and describes the probability distribution of $\Theta$ conditioned on the observed data. The likelihood is a probability distribution that assesses the probability of the data being generated, conditioned on some value of $\Theta$. Finally, the prior represents our knowledge of $\Theta$ before we observe the data $\mathbf{y}$; in this term we may include the results from previous analyses, bounds on values that $\Theta$ may take or any physically derived structure between the constituent parameters inside of $\Theta$. Prior distributions are here chosen to be uniform between $0$ and $10$ times true parameter values.

Exact computation of $\mathrm{p}(\Theta \mid \mathbf{y})$ is analytically achievable for a small class of model problems; however, this is typically not so and so $\mathrm{p}(\Theta \mid \mathbf{y})$ is computed numerically using Markov chain Monte Carlo (MCMC), as this is the gold standard in statistical computing. MCMC generates a dependent chain of draws from the posterior $\mathrm{p}(\Theta \mid \mathbf{y})$ such that subsets of $\Theta$ are visited proportionally to the posterior probability of the subsets. We show an example of this in Figure 2 for the moored data reference scenario. The traceplots consist of 20,000 dependent samples from which we may derive summaries of the

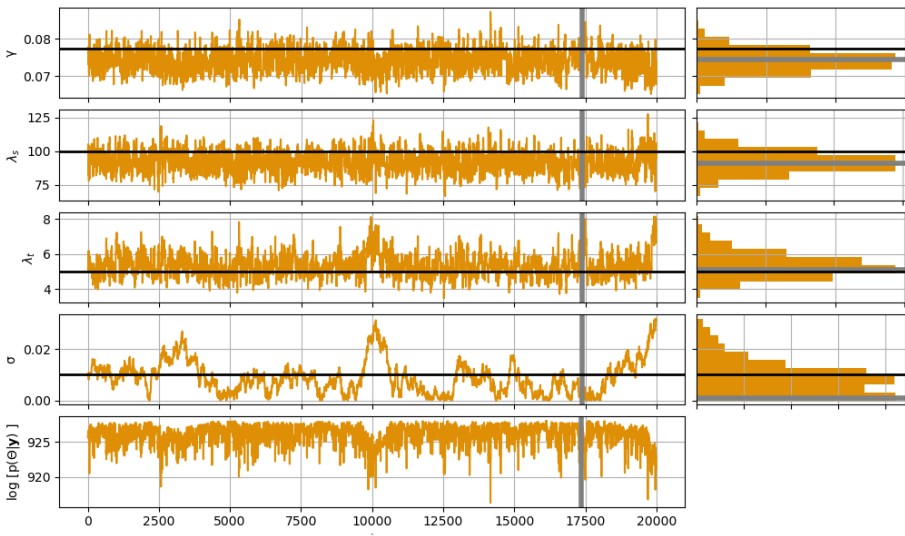

**Figure 2.** Trace plots of MCMC sampling for each flow parameters (left) and associated histograms (right) for a single inference based on 8 drifter trajectories for the reference scenario. True parameter values are indicated by the black lines, while MAP location and values are indicated by thick gray lines.

posterior distribution, $p(\Theta \mid \mathbf{y})$, via standard Monte Carlo methods. For example, the marginal distributions of each parameter are represented by the histograms in the right-hand column of Figure 2. MCMC is *asymptotically exact* in that the sampled draws converge to the exact posterior probability distribution. We generate samples using Metropolis-Hastings (MH), a well-known and accessible MCMC algorithm. Description and particulars are provided in the appendix.

As discussed above, we parameterise our model using the Matérn covariance function as it exemplifies a number of desirable
physical characteristics. However, the derivative of the Matérn covariance function is difficult to obtain due to $\mathcal{K}_\nu(\cdot)$: analytical derivatives are only available at integer values of $\nu - 1/2$, and numerical calculations of $\mathcal{K}_\nu(\cdot)$ are not available in any symbolic toolboxes that we are aware of. To mitigate the computational burden and enable the performance of ensemble of statistical experiments, we decided to fix the slope parameters $\nu_s$ and $\nu_t$ to half-integer values described in section 2.2 and exclude these parameters from those inferred. This choice is relaxed in section 3.5 in order to demonstrate that the inference of these
parameters is possible as achieved in the purely temporal domain by Sykulski et al. (2016).

The noise chosen here ($\sigma = 0.01$ m/s) is representative of oceanographic velocity observations. State of the art surface drifters are for instance equipped with GPS and provide hourly position observations with 10 m accuracy (Poulain et al., 2022). Under the assumption that the noise is white, this leads to a noise standard deviation on velocity observations of about $0.004$ m/s that is less than half of what we considered. Acoustic Doppler current profilers (ADCP), such as those deployed on
moorings, exhibit typical accuracy's in the $0.01$ to $0.1$ m/s range (Klema et al., 2020). Our expectation is that the performance of the inference should eventually decay as the noise on velocity observations increases and this should be verified in a future study.

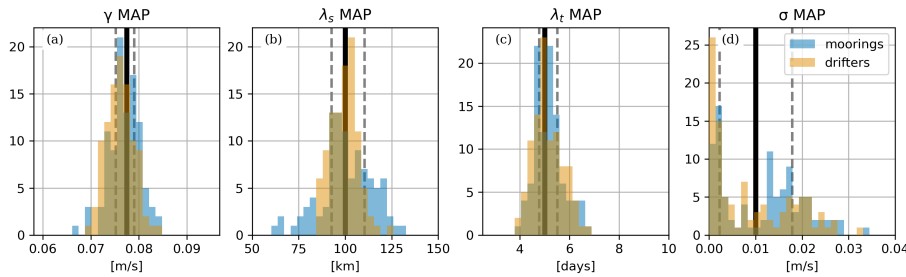

**Figure 3.** Distribution of parameters MAP values for the reference flow and reference observation scenario (scenario REF). True parameter values are represented by vertical black lines. First and third quartiles are gray dashed vertical lines and provide insight into the inter-quartile width (IQW).

## 2.4 Validation of the Inference Methodology

As the mooring and drifter data are simulated, we know the ground truth, and so may validate the MCMC sampling methodology. We show this for two cases: first, we show the probabilistic parameter estimates from the reference flow (section 2.1); and second, we compare the maximum-a-posteriori (MAP) estimates, i.e. $\hat{\Theta} := \arg\max_{\Theta}\{p(\Theta \mid \mathbf{y})\}$, of an 100-member ensemble with their true values. Examining a single scenario demonstrates the inherent uncertainty associated with a single experiment; whereas, inference across an ensemble looks at the variability that arises between data-samples. In all cases, the data comprise a bivariate $u$, $v$ time-series collected either along 8 trajectories (drifters), or at 8 stationary locations (moorings), with 2 days temporal sampling over 100 days, amounting to 400 data points. The ensemble data are generated from the single spatio-temporal field with randomly sampled drifter tracks and mooring locations. Figure 2 shows the marginal posterior probability distributions of the single-member reference scenario. For all parameters, the true values lie well within the probability distribution. Note, $\sigma^2$ is not well resolved, this is because the roughness of the Matern process, at the set sampling interval (see Figure 1), confounds with the noise signal so that both processes may be viable in explaining the observed data. This is somewhat expected and more detailed statistical diagnostics accompany the code in the supplementary material. Figure 3 plots a histogram of the MAP values calculated from each ensemble member's MCMC chain against the true value. This shows the variability of the distributions about the true value over the ensemble. Again, all distributions are centered on the true values, and there exists some difficulty in observing $\sigma^2$ with precision. The precision of the inference will also be quantified by the difference between the third and first quartiles which will be referred to as the inter-quartile width (IQW).

## 2.5 Inference scenarios

This study reports on the performance of the inference method under several scenarios (summarized in Table 1):

- REF corresponds to the nominal configuration described in Section 2.4 with 8 simultaneously deployed platforms

- SEP[$dx$] - When multiple platforms are simultaneously sampling the flow, the separation between platforms and more generally their geometrical distribution are expected to modulate the performance of the inference. To simplify the

**Table 1.** Inference scenarios. All other parameters are held constant across the scenarios.

| scenario | $\gamma$ [m/s] | $N_p$ | drifters | moorings |
|---|---|---|---|---|
| REF | $7.7 \times 10^{-2}$ | 8 | random draw | random draw |
| SEP[$dx$] | $7.7 \times 10^{-2}$ | 2 | random with initial separation $dx$ | random with separation $dx$ |
| IND[$N_p$] | $7.7 \times 10^{-2}$ | [1-16] | random draw and independent observations | random draw independent observations |
| OPT[$N_p$] | $7.7 \times 10^{-2}$ | [1-16] | spiral deployment | spiral deployment |
| REG[$\alpha$] | $[1.6 \times 10^{-3} - 4 \times 10^{-1}]$ | 1 | random draw | random draw |
| NU | $7.7 \times 10^{-2}$ | 8 | spiral deployment, $(\nu_s, \nu_t)$ inferred | spiral deployment, $(\nu_s, \nu_t)$ inferred |

analysis, we restrict the configuration to two simultaneous observing platforms (e.g. two drifters or two moorings) and investigate the sensitivity of the inference performance to their separation (with 10% tolerance). For drifters, the separation is the initial one between the two drifters.

- IND - inference is performed by assuming time series from different platforms are independent from each other. Such a situation would occur if individual moorings/drifters were deployed the same location but at times sufficiently far apart, no correlation is expected across the velocity time series recorded by each platform. In effect this amounts to quantifying the ability of one platform at capturing flow parameters and investigating the sensitivity to the length of the time series.

- OPT[$N_p$] - platforms are deployed in a spiral configuration that leads to separations that span the flow spatial decorrelation scale (see section 5.2). The purpose of this experiment is to perform a simple experimental design optimization of the number of platforms deployed and of the choice between moorings and drifters.

- REG[$\alpha$] - the amplitude of the flow is rescaled in order to explore different values of the flow regime parameter $\alpha = U\lambda_t/\lambda_s$. The amplitude of the noise is linearly scaled as a function of $\alpha$ in order to maintain a fixed signal to noise ratio. Inference are performed with a single platform.

- NU - This scenario is similar to OPT[$N_p$] with $N_p = 8$, with the exception the spectral slope parameters $\nu_s$ and $\nu_t$ are also inferred.

## 3 Results

### 3.1 Platform separation sensitivity

Under scenario SEP[$dx$], estimations of the flow amplitude are comparable for observations from two moorings or two drifters and precise with IQWs lower than 13% of true amplitudes and no sensitivity to separation (Figure 4a). We argue this follows from the fact that inferences are provided with velocity observations as inputs. Drifter inferences of the flow amplitude exhibit a 1% to 3% low bias which is comparable to that associated with turbophoresis (Section 2.2).

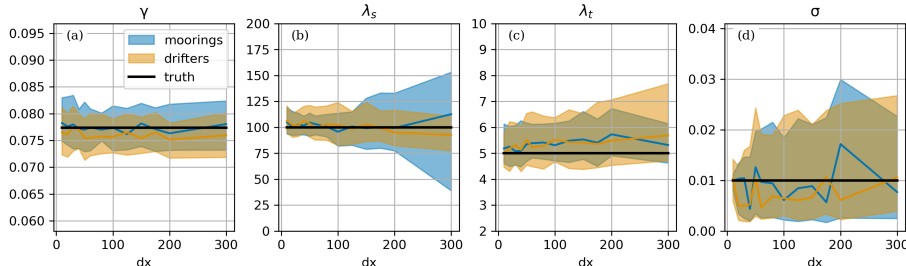

**Figure 4.** Sensitivity of parameter MAP estimates to platform separation (in km) for the 2 platform configurations (scenario SEP[$dx$]). Lines represent the median, while shaded areas are bounded by first and third quartiles. True values are in black. When visible, gray shadings represent the no-go zone of the prior and inference parameter exploration.

Mooring spatial scale estimates are sensitive to separation (Figure 4b). After a modest decrease in performance of the inference with separation as measured by IQWs, the best inference is obtained for separation in the range of 40 to 80 km. For larger separations, the inference precision decreases with IQW reaching values of about 0% of true values at 300 km, i.e. 3 times the flow spatial scale. This loss of performance with separation reflects the loss of correlation between the flow measured by each mooring and thus the lack of information about spatial structure in the dataset. Drifters exhibit no clear sensitivity to
separation for the spatial scale estimate. This may first be explained by the substantial displacements of the drifters compared to the separations considered. A flow exponentially autocorrelated over 10 days with a standard deviation of 10 cm/s leads to an absolute dispersion of $(250 \text{ km})^2$ (Gurarie et al., 2017). The natural ability of drifters to explore space and time and therefore constrain spatial scales (see section 3.2) provides a second explanation.
Mooring and drifter inferences of the flow temporal scale both exhibit a modest high bias of 5 to 10% (Figure 4c). As expected, drifters are overall less effective than moorings at estimating the flow temporal scale parameter. IQW's associated with drifter inferences are systematically larger than those associated with moorings which fluctuate around 30% for moorings compared with drifters which increase with separation up to 60%.

## 3.2 Sensitivity to the number of independent platforms

Under scenario IND[$N_p$], single moorings (i.e. $N_p = 1$) provide estimates of the flow amplitude, $\gamma$, and temporal decorrelation scale, $\lambda_t$, parameters that are precise, with IQW starting at about 16% and 45% of true values, respectively. Parameters $\gamma$ and $\lambda_t$ converge to true values as the number of independent moorings is increased (Figure 5). For the maximum number of platforms considered, the IQW of the flow amplitude and temporal decorrelation has decreased to 4% and 11%, respectively. As expected from their inability to explore the spatial dimension, single moorings are however globally unable to capture the flow spatial
scale with IQW comparable to half the width of the parameter space allowed to be explored, i.e. [0, 1000 km], which amounts to the prior uncertainty (that is, there is no resolution of uncertainty).

In comparison, drifters provide reasonable estimates of all three flow parameters ($\gamma$, $\lambda_s$, $\lambda_t$) with IQW starting at about 14%, 92%, 95% for one platform. These estimates converge toward truth as the number of platforms is increased with IQW

smaller than 16% for all three parameters with 16 drifters. The ability of drifters to capture both spatial and temporal scales is explained by their ability to sample space and time simultaneously. MAP medians indicate mild biases with an underestimation of amplitude and overestimation of temporal decorrelation scale which decrease as the number of drifters is increased. The amplitude low bias is about 7% with a single drifter and reduces to about $1.4\%$ with 16 drifters, which is comparable to the turbophoresis bias (Section 2.2). The temporal decorrelation scale $\lambda_t$ of drifters are always less accurate than that obtained with moorings which we interpret as the price to pay for the simultaneous sampling of spatial and temporal variability.

## 3.3 Experimental design optimization

Optimizing an experimental design is a complex task that results from a compromise between scientific goals, a priori knowledge of the variables to be measured, financial and logistical constraints, and the need for redundancy, among other aspects. Scenario OPT[$N_p$] illustrates how one could identify what is the minimum experimental design, enabling an accurate estimation of flow properties.

Consistent with the results of the previous scenarios, no substantial bias is observed. IQW is used to quantify accuracy and therefore is the target variable to minimize to identify optimal design (Figure 6). Apart from the one platform configuration, where the mooring is unable to estimate the spatial scale of variability, moorings and drifters present comparable sensitivities as a function of the number of platforms. The number of platforms required to reach a target IQW of 20% of the true value for all parameters except for $\sigma$, is 4 for both platforms (Figure 6).

In light of the low cost of drifters compared to moorings (factor of about 100 for deep sea applications), this result is particularly striking. However, we note the simplicity of the present exercise (idealized flow, constrained geometry of deployment, see section 5.2) in light of past efforts on the matter (Bretherton and McWilliams, 1980; Barth and Wunsch, 1990). As stated in the preamble, optimizing for characterization of flow properties constitutes one consideration among many that may be taken in an experimental design optimization. Scientific goals may in general go well beyond the characterization of flow properties. If flow properties are suspected to evolve temporally, the use of drifters which are expected to eventually disperse will require multiple deployments in the area of interest unlike with moorings.

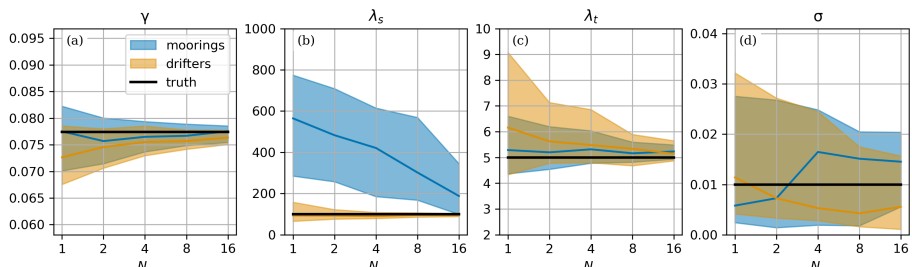

**Figure 5.** Sensitivity of parameter MAP estimates to the number of platforms (scenario IND[$N_p$]). Platforms are assumed independent from each other. Same representation as Figure (4).

## 3.4 Flow regime sensitivity

We turn now to an investigation of the sensitivity of inferences to the flow parameter $\alpha$ (scenario REG[$\alpha$]). We revert to the single platform configuration in order to limit the exchange of information across platforms and the resulting constraint it brings for inference which may mask the $\alpha$ sensitivity. For comparison purposes we also perform a "time-only" inference of drifter velocity time series which estimates flow amplitude, temporal decorrelation scale and noise only and not the spatial decorrelation scale $\lambda_s$.

As anticipated from section 3.2, inferences of flow amplitude and temporal decorrelation scale from mooring observations are relatively accurate with IQW of about 15% and 50% of true values, respectively (Figure 7a and 7c). The amplitude inference reflects the linear sensitivity to $\alpha$. Spatial scales remain undetermined for all $\alpha$ values (Figure 7b). This lack of sensitivity is expected due to exclusive sampling of temporal variability by a single mooring.

Inferences of flow amplitude from drifter observations are comparable to mooring inferences in terms of IQW albeit with a low bias of about 5% (Figure 7a). A comparable bias is observed on time-only inferences for small $\alpha$ values but is exacerbated for $\alpha$ larger than unity where it reaches about 35% of the true amplitude (Figure 7c). For large $\alpha$, distortions of the temporal spectrum shape is likely affecting the overall performance of the time-only inferences which rely on the spectral distribution following that of a Matérn $1/2$ process.

For small $\alpha$ values ($< 0.2$), inference of the flow spatial decorrelation scale from drifter observations are the worst and the IQW is nearly comparable to those from mooring observations (Figure 7b). Drifters indeed merely move over a flow time scale comparable to the spatial decorrelation scale in this flow regime, which has been historically coined a "fixed-float" and can be effectively considered a mooring (Middleton, 1985; Lumpkin et al., 2002). Accordingly, when $\alpha < 0.2$ estimates of the flow amplitude $\gamma$ and temporal decorrelation scale $\lambda_t$ are comparable for moorings and for drifters whether with the standard inference or the "time-only" inference.

For larger values of $\alpha$ (e.g. $> 0.2$), the precision of the flow spatial decorrelation scale inference from drifter observations improves substantially with decreasing IQW (down to 50% at $\alpha \sim 1$). In contrast, estimates of the temporal decorrelation scale deteriorate with a bias high of about 25% and IQW width of about 120%. At these values of $\alpha$, the flow is in the so called "frozen turbulence" regime and drifters are in effect experiencing the spatial variability of the flow field (Middleton, 1985;

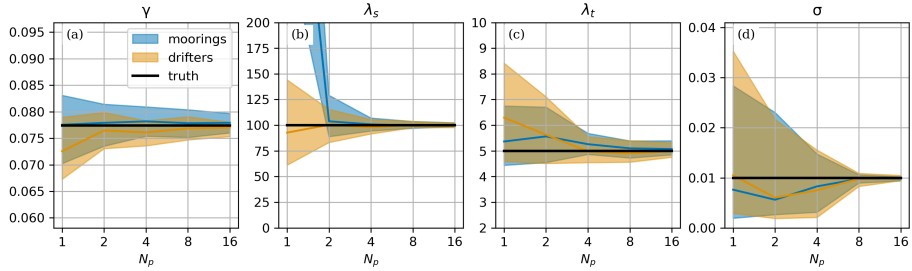

**Figure 6.** Sensitivity of parameter MAP estimates to the number of platforms (scenario OPT[$N_p$]). Same representation as Figure (4).

Lumpkin et al., 2002). This is directly reflected in the estimate of the temporal scale obtained from the "time-only" inference which monotonically decreases with $\alpha$. The fact that the temporal scale from the space-time inference does not follow a similar trend is a testimony to the relevance of the latter method which is able to identify that observations reflect a predominance of spatial variability and attribute reasonable space and time scale estimates, albeit with moderate error and bias.

### 3.5 Spectral slope estimation

For the final experiment (NU), the assumption that spectral slopes are known is relaxed and Matérn slope parameters $\nu_s$ and $\nu_t$ are inferred along with the other parameters, i.e. $\gamma$, $\lambda_s$, $\lambda_t$, $\sigma$. The assumed prior distributions are uniform over $[1,5]$ and $[0,5]$ for $\nu_s$, $\nu_t$ which is larger than typical uncertainties in the ocean about these parameters. Estimating these parameters leads to a 45-fold increase in computing time, due to computation of the Bessel function $\mathcal{K}_\nu$, as discussed in Section 2.1.

The impact on flow parameter estimation is a modest increase of normalized IQW (Figure 8) compared to OPT[$N_p$] with $N_p = 8$ (Figure 6). For instance, spatial and temporal decorrelation scales IQW estimated with mooring observations increase from 7 and 14% to 10 and 18% of true values respectively. Inferences from drifter observations undergo comparable increases.

The inference of spatial and temporal Matérn slopes are successful with posterior distributions centered around their true values, and, IQW of less than 22% of true values. Independent experiments with random platform deployments similar to REF lead to more contrasted results with the temporal slope being effectively resolved but not the spatial slope (not shown). This is is an indication that the estimation of Matérn slope parameters is more demanding on observation quality and information content. Pending improvements in the performance of the inference computation, these results present promising perspectives for the systematic inference of spectral slopes.

## 4 Conclusions

We have presented a novel Bayesian method to infer surface ocean circulation spectral parameters (e.g. amplitude, and spatial and temporal decorrelation scales) from sparse observations of the flow. The intention was to quantify parameter uncertainty due to sampling and flow regimes. These results may guide the design and analysis of future field campaigns and open

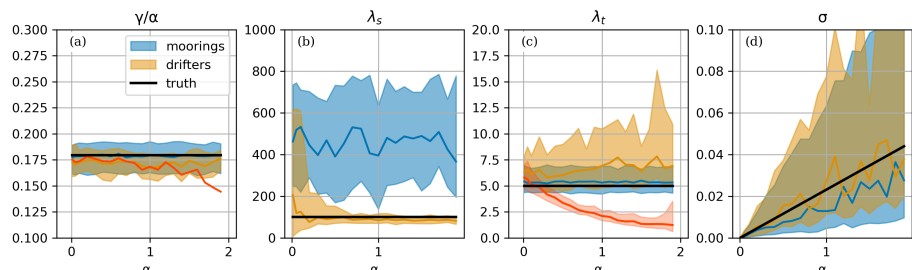

**Figure 7.** Sensitivity of parameter MAP estimates to flow regime $\alpha$ for the single platform configuration (scenario REG[$\alpha$]). Time only drifter inference is in red on (a) (median MAP dashed) and (c) (quartiles and median). Same representation as Figure (4) otherwise.

novel avenues for the analysis of existing datasets. We considered flow observation from two platforms typically employed in oceanography: moorings which provide fixed point flow observations and drifters that provide along-flow flow observations. Inference based on both types of platforms provide flow characterization estimates that converge to true values as the number of observations is increased. The performance of the method was quantified in various observing configurations which allowed us to highlight the pros and cons of each type of platform. As already recognized, moorings are well suited to characterize temporal scales of variability and if deployed as appropriately spaced simultaneous networks can constrain flow spatial scales. Drifters naturally sample both space and time and we showed they can simultaneously constrain and separate the flow's space and time scales even when deployed in isolation which is the first demonstration to our knowledge. We also showed that the ability of drifter observations to characterize flow properties depends on a non-dimensional parameter that quantifies the relative magnitude of the spatial and temporal decorrelation scales. Given the relative low cost and low environmental impact associated with drifter deployments compared with moorings, we argue they provide a powerful and more sustainable means to characterize surface flow properties. Finally, the present inference method may be of more general relevance for the simultaneous space-time characterization of flow properties in other fluid mechanics configurations (Reneuve and Chevillard, 2020).

More developments are required in order to make this method applicable to realistic oceanographic configurations. First the method needs to be extended to flows that are composed of a superposition of processes commonly occurring in the ocean, e.g. internal waves and tides, near-inertial waves. Such an extension will present methodological challenges associated with the parametrization of the space/time variability associated with these processes. The assumption of space/time separability, which was imposed here by the selected method of flow field generation, may have to be relaxed in a realistic configuration (Wortham and Wunsch, 2014; De Marez et al., 2023). As long as correlations may be expressed in physical space, extension of the inference to non-separable cases is direct. The issue introduced by non-separable kernels is arguably the most difficult challenge that needs to be addressed. It may also be useful to generalize the inference method to simultaneously account for observations that are of diverse nature, for instance current observations from drifters, pressure from moorings, sea level observations from satellite altimetry. Such an extension will require deriving the expected correlation between each of the variables concerned and will in any case depend on the process modeled. A first application of the method to real data may be with gridded altimetric sea level or current data (AVISO+). The smoothing applied to generate these products may allow alleviation of the complexity associated with high-frequency processes.

Moving to a more realistic flow configuration will require evolving the synthetic flow strategy. The present choice allowed us to generate flows with arbitrary spatiotemporal structure, including some flows that are unlikely to occur in the ocean, in order to enable a broad exploration of the inference performance. This approach could be pushed further with the superposition of multiple processes and non-separable kernels and will likely require leveraging spectral domain approaches. As highlighted in Section 3.5, there are some computational difficulties with estimating the spectral slope via the Matérn covariance function. Slope estimation in the spectral domain is simple as the slope appears in the PSD in an analytically tractable form (see Sykulski et al., 2016); however, for drifter based inference, as we are interested in estimation of the Eulerian properties, we cannot use such Lagrangian spectral techniques. There are some recent results that resolve the computational burden imparted by the cal-

culation of the Bessel function and its derivatives (Geoga et al., 2022). Regrettably, at the time of writing, code for this study's methodology is not widely available across coding platforms. We hope that this, or similar methodological advancements, may be included in future work that will focus on estimating more realistic flows. Using flows generated from dynamical models (quasi-geostrophic, primitive equations) may eventually be necessary to capture regimes of variability more closely representative of the actual ocean dynamics with more realistic representations of process life cycles. Finally, the advection of drifters could account for a stochastic component in order to represent inaccuracies associated with the dynamical system considered (Mínguez et al., 2012).

Applications of the inference method to realistic observation datasets (e.g. velocity observations from the Global Drifter Program - Lumpkin et al. (2017)) would be computationally prevented in the present form due to the use of dense covariance arrays. Alleviating this constraint will require us to leverage sparsity in the inference inputs associated with observations that are distant in space and/or time compared to associated decorrelation scales. This represents the second most important challenge that will have to be faced for applications in realistic configurations. Data collected from regional campaigns may be more suitable in the short term.

*Code availability.* The software code required to reproduce results are found at the following url: https://github.com/apatlpo/nwastats

*Video supplement.* Animation of the synthetic flow and drifter trajectories in the REF, as well the REG[0.008] and REG[1.6] scenarios are provided.

*Author contributions.* All authors contributed to the conceptualization of this study. A.P., L.A., M.R., A.Z. developed the software required to perform the analysis. A.P. and L.A. conducted the investigation. A.P. produced the visualization. A.P. and L.A. prepared the original manuscript. A.P., L.A., M.R., A.Z., N.J. reviewed and edited the manuscript. N.J. and A.P. acquired funding to make this work possible.

*Competing interests.* The authors declare that they have no conflict of interest.

*Acknowledgements.* A.P. acknowledge support from the Institute of Advanced Studies (University of Western Australian) under the Gledden Visiting Fellowship program, as well as support from the TOSCA-ROSES SWOT project DIEGO. We acknowledge support by the Australian Research Council (ARC) Industrial Transformation Research Hub for Transforming energy Infrastructure through Digital Engineering (TIDE), Grant No. IH200100009, and ARC grants LP210200613 and DP210102745.

# 5  Appendix

## 5.1  MCMC Sampling

### 5.1.1  Metropolis-Hastings Algorithm

The Markovian property of MCMC implies that a sample $\Theta^{[i]}$ only depends on its previous sample $\Theta^{[i-1]}$; the method by which $\Theta^{[i]}$ is generated from $\Theta^{[i-1]}$ distinguishes the various MCMC algorithms. All MCMC algorithms propose some $\Theta^{[*]}$ from $\Theta^{[i-1]}$ and with probability $\alpha$ either accept $\Theta^{[*]}$, in which case $\Theta^{[i]} = \Theta^{[*]}$, or reject $\Theta^{[*]}$, in which case $\Theta^{[i]} = \Theta^{[i-1]}$. The Metropolis-Hastings (MH) algorithm, initially presented in Metropolis et al. (1953) and later extended by Hastings (1970), generates a proposal $\Theta^{[*]}$ from $\Theta^{[i-1]}$ using some user specified proposal distribution $f(\Theta^{[*]} \mid \Theta^{[i-1]})$. Given a proposal $\Theta^{[*]}$, we accept the sample with probability $r$, where

$$r = \min\left(1, \frac{p(\Theta^{[*]} \mid \mathbf{y}) f(\Theta^{[i-1]} \mid \Theta^{[*]})}{p(\Theta^{[i-1]} \mid \mathbf{y}) f(\Theta^{[*]} \mid \Theta^{[i-1]})}\right). \tag{13}$$

If the proposal density is symmetrical, that is, $f(\Theta^{[i-1]} \mid \Theta^{[*]}) = f(\Theta^{[*]} \mid \Theta^{[i-1]})$, then (13) reduces to the ratio of the posterior densities and so the MH algorithm will always accept a proposed $\Theta^{[*]}$ that is more probable than $\Theta^{[i-1]}$. The choice of $f(\cdot \mid \cdot)$ is critical to the success of the MH algorithm. If $f(\cdot \mid \cdot)$ is too wide then the algorithm can become stuck for many iterations, thus generating very few unique proposals. Conversely, if $f(\cdot \mid \cdot)$ is too narrow the algorithm will not effectively explore the parameter space, the sampled $\Theta^{[1]}, \ldots, \Theta^{[n]}$ will be highly correlated, and again, few independent samples will be generated. One of the main drawbacks of the MH algorithm is that there are sampling parameters that need to be hand-tuned.

We parameterize $f(\cdot \mid \cdot)$ as a multivariate normal distribution with mean $\Theta^{[i-1]}$ and diagonal covariance matrix. A widely agreed upon rule-of-thumb to balance exploration and exploitation of the posterior distribution is an acceptance probability of $\sim 0.25$. Accordingly, we set the standard deviations of the proposal distribution to be between 0.05 and 0.2 of the true parameter values, corresponding to situations where we have larger and lower instances of observed data. The reason for this is simple: as the number of observations increases, the uncertainty of our parameter values decreases, implying a tighter posterior distribution. Consequently, a tighter proposal distribution is required to achieve a comparable acceptance probability. Full validation results to guarantee fit and convergence of the MCMC estimation algorithm are presented alongside the code at https://github.com/apatlpo/nwastats.

### 5.1.2  Notes on alternative MCMC sampling algorithms

Modern MCMC algorithms have been dominated by gradient-based proposal methods where a proposal $\Theta^{[*]}$ is generated by assessing the local topology surrounding $\Theta^{[i-1]}$: this allows the algorithm to efficiently trade off notions of exploration and exploitation of the posterior. Included in these algorithms are the popular Hamiltonian Monte Carlo techniques, such as those implemented in Stan (Carpenter et al., 2017), PyMC3 (Salvatier et al., 2016) and Pyro (Bingham et al., 2019); these implementations, as well as others such as GPJax (Pinder and Dodd, 2022) will typically use symbolic toolboxes to define the local topology of the posterior. Alternative MCMC algorithms should not affect the accuracy of the posterior estimation;

but rather, they will differ in their sampling efficiency. This study is concerned with inference, and not operationalization, and so we choose the Metropolis-Hastings algorithm so as to avoid the issue of gradients at the cost of some hand-tuning of the algorithm.

## 5.2  Platform array design

For the experiment OPT[$N_p$], platforms are deployed at locations that aim to span a wide range of platform separations around some expectation of the spatial scale decorrelation. For that purpose, locations were set along a spiral defined by its spatial footprint $L$, orientation $\beta$, and center $(x_c, y_c)$ according to:

$$x_j + iy_j = x_c + iy_c + r\theta_j \times e^{i(\theta_j + \beta)}, \text{ with} \tag{14}$$

$$\theta_j = j \times \delta, \text{ and, } r = \begin{cases} L/\delta, & \text{if } N_p = 2 \\ L/(2N_p\delta), & \text{otherwise} \end{cases} \tag{15}$$

where $0 \leq j < N_p$ is a platform digit identifier. We have made the choice $\delta = \pi/3$. An illustration of such platform deployment is illustrated in Figure 9a. The distribution of platform separations successfully spans the ensemble of length scales up to $L$ (Figure 9b).

Each draw in the OPT[$N_p$] ensemble experiment is based upon uniform random draws of the spiral center within the domain, of the spatial footprint $L$ within [50km ,300 km], and of the orientation $\beta$ within $[0, 2\pi]$.

## 6  Notations

**Table 2.** Notations

Inferred parameters:

| | |
|---|---|
| $\gamma$ | streamfunction amplitude to spatial decorrelation scale ratio |
| $\lambda_s$ | spatial decorrelation scale |
| $\lambda_t$ | temporal decorrelation scale |
| $\sigma$ | noise standard deviation |
| $\nu_s$ | spatial slope parameter, only inferred in section 3.5 |
| $\nu_t$ | temporal slope parameter, only inferred in section 3.5 |
| $\Theta$ | vector composed of all inferred parameters |

Other parameters:

| | |
|---|---|
| $U$ | flow amplitude |
| $\Psi$ | streamfunction amplitude |
| $\alpha = U\lambda_t/\lambda_s$ | non-dimensional flow parameter |
| $N_p$ | number of observing platforms (e.g. drifters or moorings) |

Variables:

| | |
|---|---|
| $x, y$ | spatial coordinate or increment |
| $t$ | temporal coordinate or increment |
| $u, v$ | horizontal velocity field |
| $\psi$ | streamfunction |
| $\phi$ | flow potential |
| $C_{ab}$ | cross-correlation between variables $a$ and $b$ |
| $S_{ab}$ | cross-spectrum between variables $a$ and $b$ |
| $(k, l)$ | horizontal wavenumbers |
| $\mathcal{K}_\nu$ | modified Bessel function of the second kind of order $\nu$ |
| $\Gamma$ | Gamma function |
| $\mathbf{y}$ | observation vector |

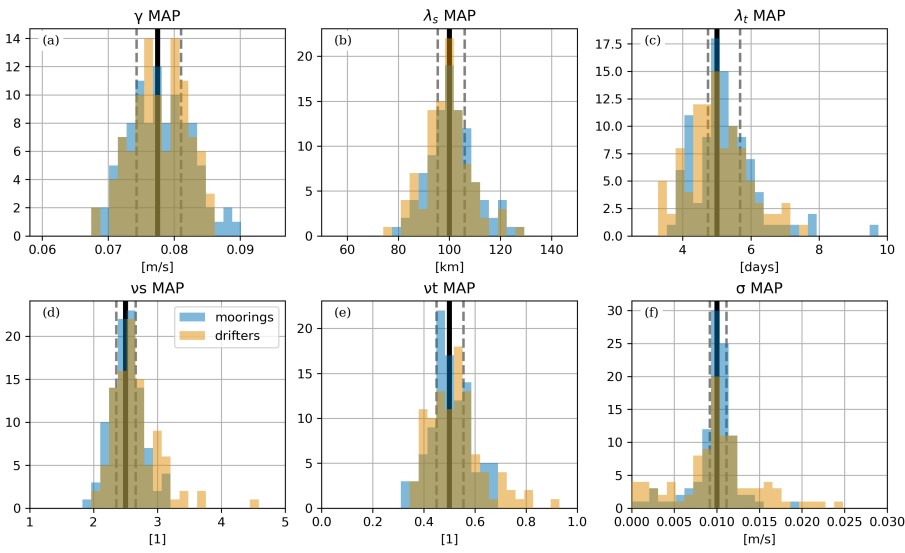

**Figure 8.** Distribution of parameter MAP values for the reference flow and reference observation scenario with inference of Matérn spectral slopes (scenario NU). True parameter values are represented by vertical black lines. First and third quartiles of the posterior distribution are gray dashed vertical lines and provide insight into IQW.

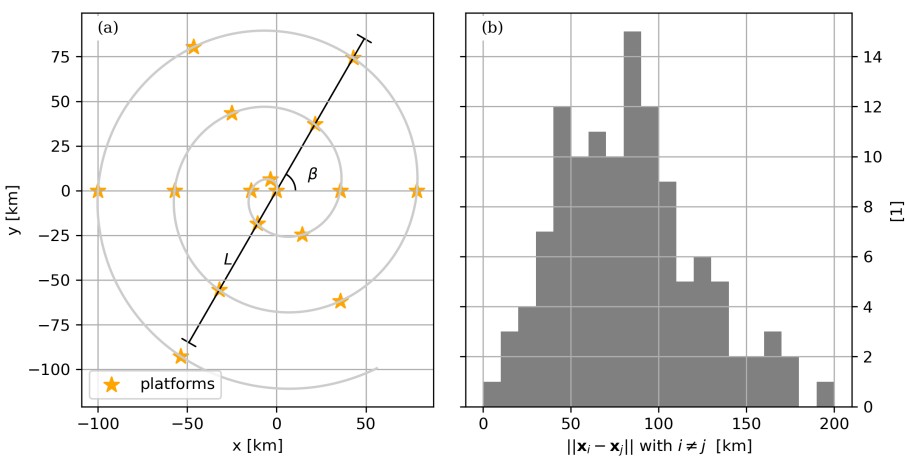

**Figure 9.** (a) Illustration of an array of $N_p = 16$ platforms for $L = 200$ km and $\beta = \pi/3$ used in OPT[$N_p$]. (b) Corresponding distribution of platform separations.

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
