# Peer review of "Inferring flow energy, space and time scales: freely-drifting vs fixed point observations"

_Nonlinear Processes in Geophysics, 2024_

## Author Response (AR1)

**"Inferring flow energy, space and time scales: freely-drifting vs fixed point observations" response to reviews**
* * *
**General response**

Dear editor and reviewers,

The present submission is composed of a revised version of the manuscript in response to reviewers' comments and questions as well as a pdf of the differences from the original manuscript and a detailed response to reviewers. Much of the modifications requested concern clarifications and justifications which we have added. The results of two more experiments (experimental design optimization, and spectral slope experiments) have been added to original ones. We conducted a modest improvement of the inference method (MCMC sampling via MH covariances) which did not change the results qualitatively but led to slightly different numbers in terms of inference performance (e.g. IQW, biases). We have also conducted a thorough editing of the language. Overall, we believe the manuscript has been substantially strengthened and we look forward to hearing about your decision regarding its publication.

Sincerely,

Aurelien Ponte & coauthors

Reviewers comments are in regular font and our responses in **bold**
* * *
RC1: 'Comment on npg-2024-10', Anonymous Referee #1, 09 May 2024

**General**:

The manuscript introduces a statistical approach to infer the spatial and temporal scales of ocean currents using drifters and moorings, under the assumption of idealized geostrophic currents. It is well-written and presents some interesting findings that offer valuable insights for optimizing future observational designs in the ocean. I anticipate further advancements and practical applications of this methodology in real-world oceanic contexts.

However, I have a few queries regarding the specifics of the geostrophic field and its applicability to real-world scenarios. While the conclusions drawn are predominantly qualitative, enhancing the experimental design and analysis could yield more quantitative results. Therefore, I have outlined both major and minor comments below for consideration.

**Major:**

One basic assumption in the surface current field is geostrophic (a purely rotational flow in line 101). The grid space is set to 2.0 km. Such a horizontal resolution, however, is able to resolve sub-meso scale motions in the ocean, which are not purely rotational.

One alternative resolution will be ~10 km under a geostrophic situation. I wonder whether a changed grid space will affect the results and final conclusion.

**Ignoring divergent motions, whether they are associated with submesoscale motions or internal gravity waves, is a choice that brings simplicity to this first study. We have listed the addition of this class of motions as a first priority in follow up studies (see second paragraph of Conclusion).**
**The grid spacing of 2 km is a factor of 50 smaller compared to the decorrelation length scale and the smallest grid spacing achievable with the computational resource at hand. A 10km grid would still be sufficient to resolve the ocean variability but may introduce more numerical errors which might add noise to the Lagrangian diagnostics. We have added further text to justify our choices in the second paragraph of section 2.2**

Also, I am interested in the practical application of the method. The AVISO products provide a near-realistic geostrophic surface current field that can be analyzed with this approach.

**Agreed, the method could be applied to AVISO surface currents and we now state this explicitly (second paragraph Conclusion) as an application along with arguments about the relevance of such exercise.**

Regional studies based on such data will have better application prospects while avoiding too much computational resources.

**Here again we agree and have added this point to the perspective section (last paragraph Conclusion)**

One important goal of this study is to provide guidance on future experimental campaigns. The conclusions are more qualitative than quantitative. Some useful quantitative information could be obtained with an improved experimental design in section 2.5. It would be better to compare the results of SEP[dx] with those in REF or just a baseline of IQW. Larger observation spacing facilitates more observation

coverage, but at the same time may lead to large IQW. Once a baseline is set, the selection of an optimal distance [dx] can be discussed. Similarly, more observations are more accurate, but will cost more. A balance can be found with a suitable number of observations.

**Many thanks for this relevant yet challenging comment. We agree the optimization of the inference was fairly qualitative in the original study. Based on the reviewer's suggestion, we have conducted one experiment labeled OPT[Np] that is more quantitative in order to optimize the experimental design for inference. The description of the results is found in new section 3.3. We are focusing on identifying two "variables" to be optimized: the number of platforms and their geometry. We found the latter to be more difficult to formalize and requires some prerequisite knowledge about scales of variability. Past studies (cited in new manuscript) have actually emphasized the difficulty around optimization of geometry. We have thus resorted to an arbitrary yet reasonable choice which is detailed in the new manuscript.**

**Minor**:

Line 2: The idealized configuration is geostrophic, of which the latter is more accurate. **We prefer the term "horizontally divergentless" instead of geostrophic as we believe it will be more directly understood by non-oceanographers**
Line 2: ingest surface velocity observation **Done**
Line 11: It's essential to consider these limitations when assessing the potential applications of any scientific method. As the author mentioned, many ocean processes, such as tides and submesoscale processes, have not been included so far, and it seems that using existing data would encounter computational challenges. **We have added "Pending further developments, " to be clear about this.**
Line 12: The spatial scale in Figure 1 seems to be meso-scale. Also, the geostrophic assumption requires meso- or large scale. **Agreed**
Line 123: what are the physical meanings of variables of U, lamda_s, lamda_t? I noticed that some of them occurred in lines 112 to 113. **This has been clarified L112, 122 and Appendix B now contains a table with all notations.**
Line 142: How was the STD of white noise estimated? Will the results changes if a larger STD value is selected? **This is now specified in the first paragraph of section 2.2 and is intended to represent moderately energetic mesoscale flows. The case of larger flow amplitudes is the topic of section 3.3.**
Figures 2 &3: I guess that the comparison between Figure 2 and Figure 3 can be used to confirm the reliability of the MAP results. However, the inconsistency in the axis ranges, especially the horizontal axis, between the two figures makes it difficult to compare them. **We have aligned the axis limits**
Subsection 2.4: Could you please go into more detail about how these parameters (i.e., U, lamda_s, lamda_t) are evaluated using u and v observations?

**We have added an illustration of the inference procedure (figure 2) as well as altering the text in Section 2.3 to be more interpretable. The technicalities of the inference algorithm have been moved to the appendix. We hope that by doing this, the nature of the computation methods are more intuitively understood.**

Line 217: I am confused about the title of section 3.2. I thought the Experiment IND[Np] was conducted to investigate the number of observations (drifters or moorings). Why the title is named related to the time series length? **We have adjusted the title of the subsection**

Figure 5: Why do moorings show better performance than drifters with smaller Np? This is also mentioned in line 236. **The last sentence of section 3.2 now provides an explanation. The better performance concerns mostly the estimation of the temporal decorrelation scale which moorings are particularly well suited to measure.**

Line 234: what dose "time-only" mean? What is the purpose of this "time-only" drifter? **We have reformulated this sentence to be clearer.**

Figure 6: the red colors are not clear enough. Also, I only found red color in fig.6c. Do the orange colors in (b) and (d) refer to the results of the "time-only" drifter? **We have fixed the labeling and used a brighter red.**
* * *
RC2: 'Comment on npg-2024-10', Anonymous Referee #2, 11 Jun 2024 reply

**General**

*The authors stochastically generate a spatially non-divergent flow field with prescribed spatial and temporal covariance functions, intended to approximate oceanic surface flow. They then consider combinations of sparse Eulerian and Lagrangian observing platforms and attempt to infer some of the parameters of the stochastic models in some of the flow regimes. The results show that, in particular, drifters capture both the spatial and temporal parameters well in some cases.*

*The strength of this manuscript is that it focuses on a fairly narrow stochastic model and a specific methodology for inferring parameters. Thus, one can assess how well the inference method is doing extracting 'true' model parameters.*

**Major comments.**

A comment on the flow model:

The authors note in the conclusion that this methodology should be extended to more realistic ocean models, and I strongly agree. The model is a sort of

two-dimensional stochastic geostrophic model, lacking even quasigeostrophic (QG) dynamics which is a foundational model for our understanding of both Eulerian and Lagrangian flow statistics. To this point, the slope parameters chosen by the authors are a -3 and -5 slope seem awfully steep compared to what we'd expect in any realistic flow; at least that is my first reaction, but I'll admit I am not certain. So either way it does seem worth justify these values and perhaps even considering QG dynamics to help the readers get a handle on where these parameters fit into known oceanic flow regimes.

**Thanks for raising this point. An unfortunate confusion amongst authors led to a typo which suggested much spectra steeper than what they actually were. We corrected the typo and took this opportunity to strengthen the justification for all parameters choices. These details are now found in the first paragraph of section 2.2.**
**The use of a statistically generated flow in place of a flow generated via a dynamical model is a choice that ensures consistency between data and inference assumptions which we believe is most appropriate for this study. We are listing the use of more realistic dynamical models in the list of future work (Conclusion, second last paragraph).**

Relatedly, the lack of waves in this model is a big deal. As noted in Beron-Vera & Lacasce's "Statistics of simulated and observed pair separations in the Gulf of Mexico", near inertial oscillations are a very dominate signal in the ocean for Lagrangian drifters (and Eulerian mooring for that matter) and fairly dramatically change the expected statistics at certain scales.

**As stated in the first paragraphs of the Introduction, our long term goal is indeed to provide improved descriptions of the full spectrum of ocean variability amongst which inertia-gravity waves represent a substantial fraction. The consideration of inertia-gravity waves is listed as a first item to be addressed in the Conclusion (second paragraph). All authors have past and on-going activities regarding the characterization and understanding of these high frequency motions. These waves do however represent a smaller fraction of the ocean surface kinetic energy compared to lower-frequency balanced turbulence (Yu et al. 2019, Arbic et al. 2022) and, for the sake of simplicity, we decided to ignore their contribution for this first thread of work and to focus on sensitivity experiments in the presence of a single lower frequency process.**

**Arbic, B.K., Elipot, S., Brasch, J.M., Menemenlis, D., Ponte, A.L., Shriver, J.F., Yu, X., Zaron, E.D., Alford, M.H., Buijsman, M.C., Abernathey, R., Garcia, D., Guan, L., Martin, P.E., Nelson, A.D., 2022. Near‑Surface Oceanic Kinetic Energy**

Distributions From Drifter Observations and Numerical Models. JGR Oceans 127, e2022JC018551. https://doi.org/10.1029/2022JC018551

Yu, X., Ponte, A.L., Elipot, S., Menemenlis, D., Zaron, E.D., Abernathey, R., 2019. Surface Kinetic Energy Distributions in the Global Oceans From a High‑Resolution Numerical Model and Surface Drifter Observations. Geophysical Research Letters 46, 9757–9766. https://doi.org/10.1029/2019GL083074

A comment on the inference method: The authors have fixed the slope of the stochastic processes a priori (as noted in the previous comment), and also fed those known slopes into their inference model. This would appear to be a fairly significant limitation to the methodology. I note that some papers have succeeded in estimating the Matern slope parameters (at continuous values) of drifter velocities using e.g. Whittle likelihood techniques such as Sykulski et al 2016, where the Matern was applied to (importantly) Lagrangian time series and not a whole spatio-temporal field with separate spatial and temporal length scale parameters. Is it the case here that the statistical inference for a spatio-temporal field (and the corresponding addition of more parameters) is the reason why the slope parameters cannot be estimated at all (with any inference method, due to data sparsity), or is this rather a computational issue of the MCMC as hinted at in the appendix A.2 where only half integer values can be efficiently used, but potentially could with another inference method? Either way the reasoning for this motivation should be discussed in the main paper and not the appendix so it's clear.

**The second interpretation is correct: calculating the derivatives of Bessel functions, as required when estimating nu at non-half-integer values, adds a difficult computational overhead to the inference. This is not an issue in the estimation of Lagrangian spectra, as in Sykulski et al. (2016), as nu appears in the PSD in a computationally tractable form. Unfortunately, we cannot replicate such inference in this problem as we are interested in estimation of the Eulerian properties whereas Sykulski et al. (2016) estimated the Lagrangian properties and so space and time (and their corresponding nu's) are aliased inside of the Lagrangian spectrum.**

**Our inference methodology does not preclude the estimation of nu, it is simply computationally demanding. We now make this point clear in Section 2.3. We have also added a single experiment that estimates nu, reported in section 3.4, and note the additional computation requirement therein.**

**There are some recent results that resolve the computational burden imparted by the calculation of the Bessel function, and its derivatives (Geoga et al., 2023). Regrettably, this study's methodology is only available in Julia and we have had troubles re-implementing the results in Python. We are in contact**

**with the authors who are working towards a C implementation, more easily executed in Python. We hope that this may be included in future work that will focus on estimating more realistic flows. We have added this discussion in the Conclusion.**

**Geoga, Christopher J., et al. "Fitting Matérn smoothness parameters using automatic differentiation." Statistics and Computing 33.2 (2023): 48.**

In addition, even if only half-integer values can be used, it is still reasonable to do some form of (possibly Bayesian) model choice over different half-integer values to select the most appropriate - perhaps the authors could investigate and apply such a procedure? At least this sort of "partial" inference allows the user to broadly select the right regime (e.g. QG with higher slopes) for the data set they have, rather than picking arbitrary and possibly wrong slope parameters which could heavily skew the other parameter estimates when misspecified significantly.

**The range of realistic slope values is unfortunately fairly small (between 2 and 5/2) and so this limits the relevance of the reviewer's suggestion. As we have mentioned in our reply above, we can estimate nu, but it is computationally demanding; and there are nascent results that may resolve this issue. As the remit of this work is to establish methodological viability, we feel that the current work is satisfactory. Questions of more realistic flows, and more expedient computation will be addressed in future work.**

**Minor comments:**

1) Please define k and l on first usage in equations (7)-(9) **Done**

2) Line 123 is unclear, which \lambda is intended, \lamba_s or \lamda_t or some combination, in the computation of the amplitude of the streamfunction? **It was \lambda_s - corrected**

3) The paper is notation heavy, a table of notation would be very helpful, including providing values of parameters that are fixed, and highlighting which ones are left for estimation. **A table was added in appendix B**

4) Please give more details on the choice of prior for the MCMC and how sensitive output is to the choice of this prior. **We specified relatively weak priors: all prior distributions were specified as uniform between 0 and 10 times the true values. In reality, these prior distributions represent extraordinarily weak assumptions on the nature of the flow (e.g. length scales are easily estimated within an order of magnitude of the true scale). Obviously, alternative prior specifications can alter the inference, but given the amount of data being fed to the MCMC inference, we expect these effects to be minor. This has now been made clear in Section 2.3.**

---

## Referee Report (RR1)

**General comments:**

Review of "Inferring flow energy, space and time scales: freely-drifting vs fixed point observations" by Aurelien Luigi Serge Ponte et al. submitted for *Nonlinear Process in Geophysics*.

The study addresses the interesting topic of the inference of spatiotemporal decomposition of oceanic surface flow variability. The authors presented a novel method to deal with the problem and evaluated the its performances in a synthetic idealized configuration. From the perspective of computation cost and environment impact, the novel method highlights the superiority of the drifter deployments.

The paper is interesting and well structured, which may have important inspirations to the scientific community. I recommend it to be accepted before some minor revisions.

More detailed comments are below.

1. The authors claimed that the observed data **y** was obtained by adding the white noise to the flow time series. The properties of white noise have important impact on the accuracy of the experiment result. Generally, different amplitudes of white noise may lead to different results. More detailed descriptions of the white noise and its possible impacts of different amplitudes are preferable in this paper.

2. The nonlinear characteristic is obvious in the oceanic surface flow variability, which means the dynamical trajectories of surface flow are sensitive to the initial conditions. Besides, the synthetic idealized configuration used in this study has no parameter errors, which are different to real oceanic systems. If the nonlinearity is taken account, whether it change the conclusion, please give some discussions.

3. The study is carried out in an ideal model, which has much differences from the real scenario. What is the most challenge, if the novel method is applied to the real oceanic systems? And how do the authors apply it to the real oceanic systems. Please clarify.

---

## Author Response (AR2)

**"Inferring flow energy, space and time scales: freely-drifting vs fixed point observations"**
**response to reviews #1**

**General response**

Dear editor and reviewers,

This novel submission is composed of a revised version of the manuscript in response to reviewer #3 comments as well as a pdf of the differences from the original manuscript and a detailed response to reviewers. Much of the modifications requested concern relatively minor clarifications. Overall, we believe the manuscript has substantially matured and we look forward to hearing about your decision regarding its publication.

Sincerely,

Aurelien Ponte & coauthors

Reviewers comments are in regular font and our responses in **bold**

'Comment on npg-2024-10', Anonymous Referee #3, 07 Sept. 2024

**General**:

The study addresses the interesting topic of the inference of spatiotemporal decomposition of oceanic surface flow variability. The authors presented a novel method to deal with the problem and evaluated the its performances in a synthetic idealized configuration. From the perspective of computation cost and environment impact, the novel method highlights the superiority of the drifter deployments.

The paper is interesting and well structured, which may have important inspirations to the scientific community. I recommend it to be accepted before some minor revisions.

More detailed comments are below.

1. The authors claimed that the observed data **y** was obtained by adding the white noise to the flow time series. The properties of white noise have important impact on the accuracy of the experiment result. Generally, different amplitudes of white noise

may lead to different results. More detailed descriptions of the white noise and its possible impacts of different amplitudes are preferable in this paper.

**We have added a paragraph (L181-L187 ) in section 2.3 that puts into context the observation noise level chosen here and lists the investigation of the robustness of inferences to noise level as a future study.**

2. The nonlinear characteristic is obvious in the oceanic surface flow variability, which means the dynamical trajectories of surface flow are sensitive to the initial conditions. Besides, the synthetic idealized configuration used in this study has no parameter errors, which are different to real oceanic systems. If the nonlinearity is taken account, whether it change the conclusion, please give some discussions.

**The flow considered in the present study is indeed extremely idealized and does not follow for instance the evolution predicted by a particular dynamical system. Moving toward more realistic configurations will require considering dynamical systems that are representative of oceanographic flows and, often as you point out, is nonlinear. The choice we have made here is justified in the first paragraph of section 2.2 and a discussion about the difficulties that will arise in more realistic configurations is found in the conclusion (second and third paragraph). Following your comment, we have added to the discussion and now mention the advection of drifters could include a stochastic component (L363-L365).**

3. The study is carried out in an ideal model, which has much differences from the real scenario. What is the most challenge, if the novel method is applied to the real oceanic systems? And how do the authors apply it to the real oceanic systems. Please clarify.

**We agree the flow considered here is extremely simplified (see reply to previous comment). The most substantial challenges regarding applications to realistic configurations are listed in the conclusion and are the following: space-time non-separable kernels and computational burden. We now explicitly mention that these are the most important challenges associated with the space-time separability L344 and L369.**